# Chosen Antioxidant Enzymes GPx4 and GPx8 in Human Colorectal Carcinoma: Study of the Slovak Population

**DOI:** 10.3390/medicina58020298

**Published:** 2022-02-16

**Authors:** Andriana Pavliuk-Karachevtseva, Jozef Mihalik, Róbert Biel, Silvia Rybárová, Ingrid Hodorová

**Affiliations:** 1Department of Anatomy, Medical Faculty, Šafárik University, Šrobárova 2, 041 83 Košice, Slovakia; andriana.pavliuk-karachevtseva@upjs.sk (A.P.-K.); jozef.mihalik@upjs.sk (J.M.); silvia.rybarova@upjs.sk (S.R.); 2Department of Clinical Oncology 2, East-Slovakian Oncological Institute, Rastislavova 43, 041 91 Košice, Slovakia; robert.biel@upjs.sk

**Keywords:** cancer, glutathione peroxidase, large intestine, antioxidants

## Abstract

*Background and Objectives:* Nowadays colorectal carcinoma (CRC) is one of the most common causes of death in patients with malignant neoplasms worldwide. Our work aimed to determine the possible involvement of glutathione peroxidases 4 and 8 (GPx4 and GPx8) in this specific tumor process. *Materials and Methods*: The expression of GPx4 and GPx8 in 58 specimens of human colorectal cancer tissues and normal tissues was detected by the indirect immunohistochemical method under a light microscope. Statistical analysis was done by Chi-squared test. Histological findings were compared with data such as gender, age, tumor grade, histotype and lymph nodes alteration. *Results:* In all specimens of healthy tissue the presence of both, GPx4 and GPx8, was detected in the cytoplasm of epithelial cells. On the other hand, a positive immunohistochemical reaction against GPx4 only in 41.4% and against GPx8 only in 29.3% of human colorectal adenocarcinoma specimens were observed. Any significant difference between the presence of GPx and the age, the gender of the patient, tumor grade, histotype of cancer and the lesion of regional lymph nodes has not been detected. *Conclusions*: Our foundation could mean, that GPx4 and GPx8 have no important role in CRC pathogenesis, but the loss of these enzymes probably indicates a serious pathological process ongoing in the large intestine. To our knowledge, this is the first paper describing GPx8 presence in human colorectal carcinoma.

## 1. Introduction

Colorectal carcinoma is a frequent cause of death in patients with malignant neoplasms. It is the second-most-occurring malignant neoplasm that causes the death of female patients, and the third-most involving male patients [1]. Colorectal cancer may be provoked by inflammatory bowel diseases such as Crohn’s disease and ulcerative colitis, or by higher age, lifestyle, smoking, excessive use of alcohol or bad eating habits [2]. However, it has recently been found that oxidative stress (OS) may also increase the likelihood of cancer [3]. OS is an imbalance between the production of reactive oxygen species (ROS) and the activity of the protective antioxidant system of the organism [3,4]. OS development leads to the disorganization of the cell structure and a change in their functional activity, leading to the cell’s death. Oxidative stress occurs when the concentration of ROS is not controlled by enzymatic (superoxide dismutases, catalase, peroxidases) and non-enzymatic antioxidants (tocopherols, ascorbic acid). Proteins, lipids and DNA may be subject to oxidative damage [5]. ROS affects the etiology and the development of several cancer forms [3]. Cancer cells show a higher acceleration in metabolic processes and require the ROS supply to maintain a high-intensity level of proliferation. Thus, ROS stimulates metabolism, invasion and metastatic spreading through gene mutation, DNA damage and protein conformation transition [3,6]. On the other hand, superoxide dismutases and glutathione peroxidases are the major enzymes catalyzing ROS decomposition reactions. These enzymes contribute to preventing oxidative damage of the organism [7]. GPxs are a group of enzymes that catalyze the reduction of H_2_O_2_ and organic hydroperoxides to water and corresponding alcohols, respectively. Up to this date, this group includes eight identified representatives, GPx1–8. In our work, we focused on two of them, GPx4 and GPx8.

GPx4 is actively studied as an enzyme that plays a significant role in ferroptosis [8]. Its importance in neurodegenerative processes [9], fertility [10] and apoptosis [11] is open to further study. It is an intracellular monomeric selenium-containing enzyme, which exists in three isoforms: cytosolic (cGPx4), mitochondrial (mGPx4) and sperm nuclear (snGPx4) [11]. The GPx4 gene is encoded in the locus of the 19p13.3 chromosome [12]. This enzyme reduces peroxidized phospholipids (PLOOHs), which are produced in membranes, and transforms them into decomposition products [13]. The enzyme inhibits cyclooxygenase (COX) and lipoxygenase (LOX) by decreasing the levels of cellular lipid hydroperoxide [14]. The importance of GPx4 in ferroptosis is also highlighted [8,13,15]. Ferroptosis is a non-apoptotic type of cell death defined by iron-dependent lipid peroxidation [15], with validity for such pathological statuses as ischemic (or reperfusion) injuries, neurodegeneration and cancer [16,17]. Ferroptotic cells store the specific lipid peroxidation products (phospholipids) [17]. In this case, the antioxidants, including GPxs, are considered as the solution to preventing ferroptosis and related diseases [15]. The fact that GPx4 is described as the ancestor of GPx8 is also important [18].

Today, GPx8 is the least known member of the glutathione peroxidases family. GPx8 is a mammalian heteromeric cysteine-rich glutathione peroxidase anchored in the endoplasmic reticulum membrane. From here, it prevents the spill of H_2_O_2_ into the cytoplasm, thereby controlling the cell redox status [19]_._ The GPx8 gene is encoded in the locus of the 5q11 chromosome [12]. It lacks a resolving cysteine in the cysteine block [11]. GPx8 is also referred to as the lung-abundant enzyme. In this case, it resembles GPx3. The enzyme increases during influenza pneumonia and decreases when the tissue starts to recover [11]. Some research also found that GPx8 is enriched in mitochondria-associated membranes, where it regulates Ca^2+^ storage and fluctuation [6]. Induction of the GPx8 expression by hypoxia-inducible factors (HIF) reduces proliferative signaling during hypoxia and/or receptor tyrosine kinases (RTK)—signaling, which contributes to oncogenesis [20,21]. However, the role of GPx8 in cancerogenesis and metastases formation is still unknown.

Our work aimed to determine the possible involvement of the chosen glutathione peroxidases in the specific tumor process. Insufficient research on the relationship between GPx8 and human colorectal carcinoma prompted us to start an investigation into this area.

## 2. Materials and Methods

### 2.1. Samples and Experimental Groups

In our studies, we used the three-step indirect immunohistochemical method of GPx4 and GPx8 detection in 58 specimens of the human colorectal adenocarcinoma. The samples were obtained from the Institute of Pathology, Louis Pasteur University Hospital Košice, Slovak Republic. A total of 37 specimens belonged to males and 21 belonged to females. The age distribution of patients was as follows: 9 of them were younger than 60 years, and 49 patients were over 60 years (Table 1). Invasion into the regional lymphatic nodes of 18 patients was found. In 6 patients, metastasis in internal organs (the liver, omentum, peritoneum and ovary) were diagnosed. Concerning the other patients, we did not acquire any information about metastases in the course of our research. Seven patients suffered from mucinous adenocarcinoma, and the rest from adenocarcinoma. A total of 8 patients were positive on the KRAS (Kirsten Rat Sarcoma virus) oncogene and 4 were negative, but analysis was not completed for 46 patients. Tumor stage was determined according to the TNM Classification system of the International Union Against Cancer. This classification includes T (describing the size of the primary tumor and whether it has invaded nearby tissue), N (describing the nearest lymph nodes that are affected) and M (describing distant metastasis).

### 2.2. Immunohistochemical Detection of GPx4 and GPx8

The paraffin-embedded sections of the human CRC were deparaffinized with xylene and rehydrated in degreasing ethanol. Finally, the 4μm slides were washed in phosphate-buffered saline (PBS) containing 0.05% Tween-20, pH 7.6. Endogen peroxidase activity was prevented by 0.3% H_2_O_2_ in 75 mL of methanol by being left in a room temperature (RT) environment for 30 min. To re-establish an original conformation of epitopes modified by fixation, antigen retrieval using a microwave was performed. The next step was the blockage of nonspecific staining with milk buffer (5% dry milk in TRIS buffer) for 30 min at RT. Primary antibodies were applied overnight in a humidified chamber at 4 °C. After rinsing the antibodies in PBS-Tween 3 × 5 min, the sections were subsequently incubated with biotinylated link (#K0675, Agilent Dako, Santa Clara, CA, USA). After, slides were then rinsed with a wash buffer, and then Streptavidin-HRP was applied (#K0675, Agilent Dako, Santa Clara, CA, USA). The presence of the enzymes was visualized by DAB (3,3′-diaminobenzidine tetrahydrochloride, #K5207, Agilent Dako, Santa Clara, CA, USA) at the concentration of 0.5 mg/mL in TRIS buffer, pH 7.6 and 0.015% H_2_O_2_. Slides were stream-rinsed with tap water, counterstained with Mayer’s hematoxylin for 1.5 min, washed again in the tap water, dried, mounted and coverslipped. The sections processed with the omission of the primary antibody served as the control group. Analyzing the negative control group showed the absence of GPx4- and GPx8-positive structures.

In the control group, we used specimens from five human colons with healthy tissue. For immunohistochemical detection, the following antibodies were exploited: rabbit polyclonal antibody for GPx4 (#bs-3884R, Bioss, Boston, MA, USA, dilution 1:150), rabbit polyclonal antibody for GPx8 (#orb183909, Biorbyt Ltd. Cambridge, UK, dilution 1:200) and rabbit polyclonal anti-GPx8 antibody (#16846-1-AP, ProteinTech, Rosemont, IL, USA). Two antibodies from different manufacturers were employed against GPx8 to verify our results concerning the GPx8 presence in the blood plasma.

### 2.3. Immunohistochemical Evaluation

For the immunohistochemical evaluation, we used quantitative methods employing Image J Software accompanied by IHC profiler Plugin [22]. Images from random fields were captured at 40× magnification using a camera (Leica ICC50 HD) attached to a light microscope (Leica DM500), and software determined the score quantitatively as no positivity (−), weak positivity (+), intermediate positivity (++) or strong positivity (+++). Samples marked as − and + were considered negative, and samples marked as ++ and +++ were considered positive. The method was the same as described previously [23].

### 2.4. Statistical Analysis

The chi-square test was employed to detect differences between corresponding statistical groups. A p-value higher than 0.05 (*p* > 0.05) was considered to be non-significant.

## 3. Results

In the healthy colon tissue, the specimens stained against GPx4 (Figure 1a) and GPx8 (Figure 1b) showed strong immunoreactivity in the cylindrical epithelium and basal cells of the colon mucosa, in all cases (100%). The strong expression (+++) of both GPxs in the regenerative cells of colonic crypts was observed. Moreover, only GPx8 was detected in the macrophages of the lymphatic follicles of the submucosa (Figure 2a) and in the blood plasma, while erythrocytes were free of the enzyme (Figure 2b).

In the pathologically changed tissue, the specimens stained against GPx4 showed cytoplasmic positivity in the cells of the colon mucosal layer in 24 cases (4 high positives and 20 positive samples), which represent 41.40%. Positive samples showed the diffuse expression of the enzyme in the columnar epithelium of the pathologically changed crypt (Figure 3a). A negative immunological reaction (Figure 3b) was observed in 34 cases (15 low positive and 19 negative samples), which represent 58.60% (Table 2).

The immunohistochemical staining against GPx8 showed high cytoplasmic positivity in the cells of the colon mucosal layer in 17 cases (4 highly positive and 13 positive samples), which represent 29.31%. Positive samples showed the diffused expressions of GPx8 in the columnar epithelium of pathologically changed crypts (Figure 4a). A negative immunohistochemical reaction (Figure 4b) was observed in 41 cases (15 low positive samples and 26 negative samples), which represent 70.69% (Table 1).

Statistical analysis did not reveal any significant differences in both the GPx4 and GPx8 (Table 2, Table 3, Table 4, Table 5, Table 6 and Table 7) groups with regard to the quantity of expression, age, gender, tumor grade, histotype or regional lymph node lesion.

## 4. Discussion

In our work, we found that both GPx4 and GPx8 are present in the epithelial cells of the healthy large intestine. In the pathologically changed tissue of colorectal cancer, the incidence of these enzymes decreases in more than half of all cases. The possible reason for GPx4 presence in colon epithelial cells is the protection of these cells, especially membranes, from ROS produced in the chymus or inside the cytoplasm. On the other hand, GPx8 in the endoplasmic reticulum is probably required for the proper folding of proteins produced by epithelial cells, such as integrins. Integrins are known as transmembrane glycoprotein receptors, which facilitate cell-to-cell adhesion and cell migration [24].

Reactive oxygen species (ROS) could initiate carcinogenesis in two ways: directly, by processes of the oxidation, nitration and halogenation of DNA, RNA and lipids, or by affecting the signaling pathways activated by them [25]. It is well known that in chronic diseases of the gastrointestinal tract, an excessive number of ROS are formed [26,27]. In turn, their high accumulation induces antioxidant mechanisms to reduce the harmful effects of ROS [27]. The redox imbalance and antioxidant activity contribute to the development of colorectal cancer. ROS are responsible for genetic alterations in colorectal cancer carcinogenesis [3]. They cause single-chain and double-chain DNA breaks and common genetic mutations, including p53, KRAS, APC and BRAF [28]. Endogenous DNA damage caused by ROS also contributes to the excessive formation of 8-hydroxy-2′-deoxyguanosine, the level of which is higher in colorectal tumors [29]. Other studies indicated a significant increase in lipid peroxidation products, such as malondialdehyde (MDA) and 4-hydroxy-2-nonenal (4-HNE) in primary colorectal cancer [30,31].

Concerning glutathione peroxidases, the decreasing GPxs’ plasma activity in patients with different tumor stages of CRC, in contrast to that of healthy people, should be noted [26]. The study on the GPx level in healthy colon tissue and CRC cases revealed a high activity of GPx in both groups. At the same time, CRC cases with serosal involvement or lymph node lesions in the affected tissue did not show the accumulation of these enzymes [32]. Another study found increased activity of the superoxide dismutase and GPx in homogenates from the CRC tumor [7].

Some studies demonstrated a high nuclear and cytoplasmic GPx4 expression in the tissue of colorectal carcinoma, as well as its high mRNA expression [33]. Three single-nucleotide polymorphisms (SNP) were found linked to an altered risk: rs713041 (GPx4), rs7579 (SEPP1) and rs34713741 (SELS) [34]. Moreover, low selenium (Se) status plays an important role in increasing the risk of CRC [35,36]. On the other hand, SNP (rs3746165) in GPx4 is linked to the risk of lethal prostate cancer [37]. Lipid peroxidases inhibit GPx4 and stimulate pancreatic cancer development [38]. The combination of sub-optimal Se status and, particularly, the genotype at the GPx4 SNP increase susceptibility to CRC, but not the risk of developing adenomatous polyps [39]. GPx4 contains Se in the form of the rare amino acid selenocysteine. This may indicate that reduced Se availability could cause a decrease in functional antioxidant GPx4 enzymes in the organism.

Currently, GPx8 is still a little-studied object of research, and its role is poorly understood. Induction of the GPx8 expression by hypoxia-inducible factors (HIF) reduces proliferative signaling during hypoxia and/or receptor tyrosine kinases (RTK) signaling, which contributes to oncogenesis [20]. The correlation between the high expression of GPx8 and a patient’s overall survival was found in those with gastric cancer and breast cancer. Patients with higher GPx8 had a worse prognosis [39,40].

In our study, we found that GPx8 is present in the macrophages of the lymphatic follicles in the submucosal layer of healthy colon tissue. On the other hand, in lymphatic infiltrations present in the affected tissue, the GPx8 was absent. Macrophages are giant phagocytes [41], the most numerous family of leucocytes in the colon, divided into two groups: proinflammatory (M1, tumoricidal) and anti-inflammatory (M2, protumorigenic) [42]. As an important part of the tumor microenvironment [42,43], tumor-associated macrophages (TAM) were mostly present in the anti-inflammatory group. Due to their anti-inflammatory and healing properties, TAM act as the suppressor of the immune response against cancer cells, promoting their growth and spread. At the same time, proinflammatory macrophages have a dual effect: they can stimulate oncogenesis in conditions of chronic inflammation due to long-term secretion of proinflammatory mediators and, conversely, stimulate the anticancer immune response and suspend the growth of tumors [42,44]. Proinflammatory macrophages also act as a source of ROS and nitrogen species or their derivatives and stimulate cancerogenesis in this way [45]. Macrophages with a GPx8 deficit contribute to the dextran-sulfate-sodium (DSS)-induced colitis phenotype. The pattern of GPx8 protein expression in different immune cell populations assumed that GPx8 was primarily expressed in macrophages, but they were not found in T or B lymphocytes or dendritic cells [46].

In patients with ulcerative colitis, the level of GPx8 in the macrophages of their colon tissue was reduced, and the level of caspase-4/11 was increased. In this case, one could argue that GPx8 acts as a negative regulator; it down-regulates caspase-4 activation and prevents colitis by decreasing caspase-4/11 [46].

An intriguing moment was the detection of GPx8 in the plasma of the specimens of healthy tissue. A similar result was found during the immunohistochemical staining of rat female genital organs [47]. This could be related to the oxidative processes in the plasma [48]. The relationship between protein disulfide isomerase (PDI) and GPx8 should be explained. The function of PDI is the isomerization of disulfide bonds in nascent ER proteins and retrograde cytoplasmic transportation during the degradation of the ER protein. GPx8 acts as the transmembrane peroxidase of PDI. PDI-rich plasma is discoverable in the healthy population, differentially expressing proteins related to cell-differentiation, protein-processing, and cell-holding functions [49]. So, as the presence of GPx8 in the plasma inside the cross-section of vessels in specimens of healthy tissue was found in our research, in the future, this point could be explored as the prognostic marker of the beginning stages of inflammatory diseases or for colon cancer risk assessment.

Performing statistical calculations, we did not find any relationships between the accumulation of antioxidants GPx4 and GPx8 in the pathologically altered and healthy tissue of the colon. This could mean that these enzymes do not play an important role in colon cancer pathogenesis. Based on our results, as well as based on other works about GPx4 and GPx8, one may assume that we could not designate these enzymes as the new prognostic markers of disease detection in the colon [50]. After all, in all healthy tissues, we found the presence of both enzymes, which were missing in the majority of pathologically altered colon specimens. This could mean that the absence of these enzymes is likely to indicate a serious pathological process in the colon. From this point of view, GPx8 is even more important than GPx4, which is absent in the mucosa of almost 71% of cases of colon tumors.

## 5. Conclusions

In our study, the presence of GPx4 and GPx8 in healthy colons and colorectal carcinoma was estimated. No statistical differences between GPx expression and the patient’s age, gender, tumor grade, histotype or the degree of invasion into their lymph nodes were found. GPx4 and GPx8 probably do not play an important role in CRC pathogenesis, but antioxidants are discussed as dual actors in tumor development. For this reason, it is important to continue using more methods in our work, which can help us to find a possible relationship between colorectal carcinoma, oxidative stress and GPxs. The presence of GPx8 in plasma also needs deeper analysis. This is the reason why, in our work, we used two different antibodies against GPx8 from two different manufacturers, with the same results. It is also necessary to take into account the fact that our work has limitations because it used only one method, immunohistochemistry. The performing of real-time PCR and immunoblotting was not possible since our research was a retrospective study and no fluids, such as plasma or blood, were available. Nevertheless, we will use them in our next experiments. However, to our knowledge, this is one of the first studies analyzing GPx8 presence in human colorectal carcinoma.

## Figures and Tables

**Figure 1 medicina-58-00298-f001:**
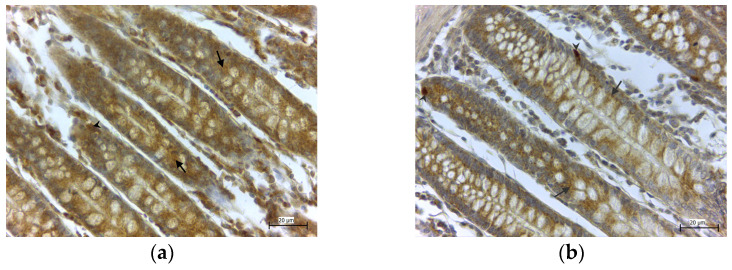
High expression of GPx4 (**a**) and GPx8 (**b**) in the columnar epithelium (arrows) and basal cells (arrowheads) in the healthy tissue of the colon.

**Figure 2 medicina-58-00298-f002:**
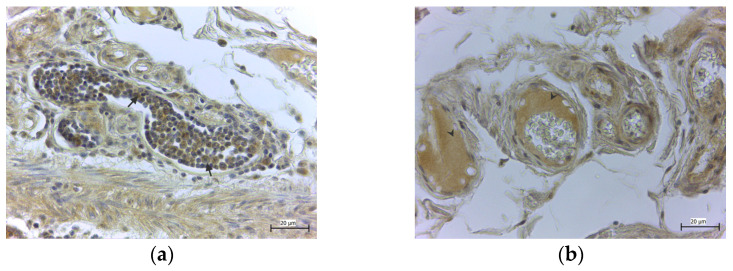
The lymphatic follicle in the submucosal layer of the healthy tissue. High expression of GPx8 in macrophages (arrows) (**a**), and positive expression of GPx8 in the plasma (arrowheads) inside blood vessels (**b**), while erythrocytes are free of the enzyme.

**Figure 3 medicina-58-00298-f003:**
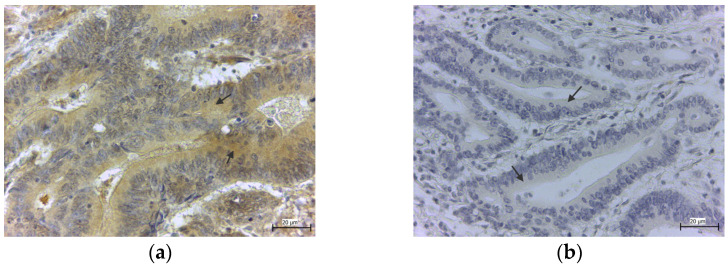
Positive (**a**) and negative (**b**) GPx4 expression in epitheliocytes of colon adenocarcinoma in pathologically changed crypts of the colon. Goblet cells in crypts are displaced by epitheliocytes (arrows).

**Figure 4 medicina-58-00298-f004:**
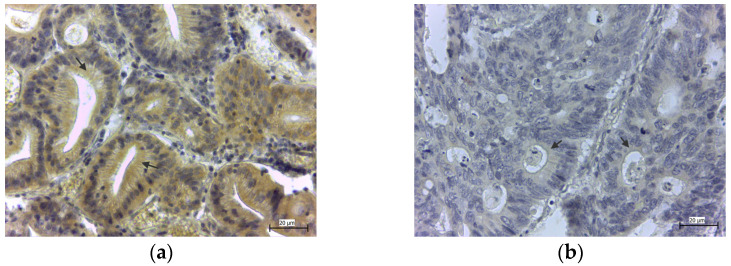
Positive (**a**) and negative (**b**) GPx8 expression in epitheliocytes of colon adenocarcinoma in pathologically changed crypts of the colon. Goblet cells in crypts are displaced by epitheliocytes (arrows).

**Table 1 medicina-58-00298-t001:** Tumor characteristics of patients.

Characteristics	No.
All patients	58
Male	35
Female	23
Age ≤60 years	10
60–70 years	25
≥70 years	23
Tumor grade I	37
II	8
III	2
Unknown	11
KRAS +	8
−	4
Analysis has not been completed	46
Mucinous adenocarcinoma	7
Adenocarcinoma	51
Invasion into lymph nodes	
Diagnosed	18
Not found or absent	44

**Table 2 medicina-58-00298-t002:** The number of GPx4- and GPx8-positive samples with the corresponding quantity of expression.

Quantity of Expression	GPx4	%	GPx8	%
+++	4	6.89	4	6.89
++	20	34.48	13	22.41
+	15	25.86	15	25.86
−	19	32.77	26	44.84
Number of positive samples	24	41.4	17	29.3
Number of negative samples	34	58.6	41	70.7

**Table 3 medicina-58-00298-t003:** The number of samples with positive and negative GPx4 and GPx8 expressions according to tumor grade.

	Tumor Grade I	Tumor Grade II	Tumor Grade III	Unknown	Fisher’s Test
GPx4 +	14	2	1	7	*p* > 0.05 (*p* = 0.22)
GPx4 −	21	8	1	4	
GPx8 +	16	3	1	5	*p* > 0.05 (*p* = 1.00)
GPx8 −	21	4	2	6	

**Table 4 medicina-58-00298-t004:** The distribution of the GPx4 and GPx8 samples according to histotype of adenocarcinoma.

	Adenocarcinoma	Mucinous Adenocarcinoma	Fisher’s Test
GPx4 +	19	5	*p* > 0.05 (*p* = 0.11)
GPx4 −	32	2	
GPx8 +	16	1	*p* > 0.05 (*p* = 0.66)
GPx8 −	35	6	

**Table 5 medicina-58-00298-t005:** The distribution of the GPx4 and GPx8 samples according to the gender of patients.

	Males	Females	Fisher’s Test
GPx4 +	14	10	*p* > 0.05 (*p* = 0.78)
GPx4 −	22	12	
GPx8 +	11	6	*p* > 0.05 (*p* = 1.00)
GPx8 −	26	15	

**Table 6 medicina-58-00298-t006:** The distribution of the GPx4 and GPx8 samples according to the age of patients.

	<60 Years	60–70 Years	>70 Years	Fisher’s Test
GPx4 +	5	10	9	*p* > 0.05 (*p* = 0.88)
GPx4 −	5	15	14	
GPx8 +	2	8	7	*p* > 0.05 (*p* = 0.86)
GPx8 −	8	17	16	

**Table 7 medicina-58-00298-t007:** The distribution of the GPx4 and GPx8 samples according to the invasion into lymph nodes.

	Invasion into Lymph Nodes +	Invasion into Lymph Nodes −	Fisher’s Test
GPx4 +	5	19	*p* > 0.05 (*p* = 0.25)
GPx4 −	13	21	
GPx8 +	8	9	*p* > 0.05 (*p* = 0.12)
GPx8 −	10	31	

## Data Availability

All employed data are included in our manuscript.

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
