# Peer review of "Chosen Antioxidant Enzymes GPx4 and GPx8 in Human Colorectal Carcinoma: Study of the Slovak Population"

_medicina, 2022, doi:10.3390/medicina58020298_

Round 1
Reviewer 1 Report
Recently I revised the manuscript entitled „ Chosen antioxidant enzymes in human colorectal carcinoma: a study of Slovak population“ for Medicina.
The manuscript needs some corrections, list below:
The title of the manuscript must be more specific, with concrete names on enzymes.
Uniform all references in the reference list, according to instruction for authors. There are a lot of mistakes.
For example, names of Journal, choose the full name or abbreviation; with or without spaces after the sign : ; lowercase or uppercase letters in name of paper, reference number 26 with underline letter Ć; etc.
Author Response
Reviewer # 1:
All changes made in the manuscript are highlighted in blue.
Question 1: The title of the manuscript must be more specific, with concrete names on enzymes.
Answer 1: Names of demanded proteins were included into manuscript title.
Question 2: Uniform all references in the reference list, according to instruction for authors. There are a lot of mistakes.
Answer 2: All references were checked and corrected according to Instructions for authors.

Reviewer 2 Report
The manuscript entitled “Chosen antioxidant enzymes in human colorectal carcinoma: a 2 study of Slovak population” is quite interesting as authors has shown the directs relationship of glutathione peroxidases 4 and 8 (GPx4 and GPx8) in clinical sample of colon cancer. The manuscript is well written, but only drawn the conclusion with limited techniques Immunohistochemical detection. Author should validate the data with real time PCR and immunoblotting techniques.
Author Response
Reviewer # 2:
All changes made in the manuscript are highlighted in blue.
Question 1: Author should validate the data with real time PCR and immunoblotting techniques.
Answer 1: Performing of real time PCR and immunoblotting techniques was not possible since our research was retrospective study and no fluids such as plasma or blood were available. Nevertheless, we would like to thank you for excellent idea, which for sure we will use in our next research.

Reviewer 3 Report
In the present work the authors invistigated the expression of GPx4 and GPx8 in CRC specimens by IHC method. The study shows for the first time the presence of GPx8 in CRC although the study was conducted on a small case series of samples.
1) In materials and methods the authors must add: the thickness of the sections and the final concentrations/dilutions of primary antibodies that they used.
2) For statistical analysis: they could apply the p value FIsher's exact test when at least one of frequency is ≤5.
3) Regarding the results, the authors must improve the quality of IHC images and add the scale bar to all IHC images
Author Response
Reviewer # 3:
All changes made in the manuscript are highlighted in blue.
Question 1: In materials and methods the authors must add: the thickness of the sections and the final concentrations/dilutions of primary antibodies that they used.
Answer 1: Demanded information was included into manuscript, more precisely the thickness of the sections was 4μm and dilution of primary antibodies was as follows: antiGPx4 1:150, and antiGPx8 1:200.
Question 2: For statistical analysis: they could apply the p value FIsher's exact test when at least one of frequency is ≤5.
Answer 2: Fisher’s test was included instead of chi square test, but the results are still not significant.
Question 3. Regarding the results, the authors must improve the quality of IHC images and add the scale bar to all IHC images
Answer 3. All old figures were replaced. New figures are in higher quality and include scale bars, too.
